# Association between sociodemographic factors, clinic characteristics and mental health screening rates in primary care

Frank Müller[1,2,3]*, Alyssa M. Abdelnour[1], Diana N. Rutaremara[1], Judith E. Arnetz[1], Eric D. Achtyes[4], Omayma Alshaarawy[5‡], Harland T. Holman[1,2‡]

1 Department of Family Medicine, College of Human Medicine, Michigan State University, Grand Rapids, MI, United States of America, 2 Spectrum Health Family Medicine Clinic, Grand Rapids, MI, United States of America, 3 Department of General Practice, University Medical Center Göttingen, Göttingen, Germany, 4 Western Michigan University Homer Stryker M.D. School of Medicine, Kalamazoo, MI, United States of America, 5 Department of Family Medicine, College of Human Medicine, Michigan State University, East Lansing, MI, United States of America

‡ OA and HTH are shared senior authorship on this work.
* muell313@msu.edu

## Abstract

### Background

Screening for mental health problems has been shown to be effective to detect depression and initiate treatment in primary care. Current guidelines recommend periodic screening for depression and anxiety. This study examines the association of patient sociodemographic factors and clinic characteristics on mental health screening in primary care.

### Design

In this retrospective cohort study, electronic medical record (EMR) data from a 14-month period from 10/15/2021 to 12/14/2022 were analyzed. Data were retrieved from 18 primary care clinics from the Corewell Health healthcare system in West Michigan. The main outcome was documentation of any Patient Health Questionnaire (PHQ-4/PHQ-9/GAD-7) screening in the EMR within the 14-month period at patient level. General linear regression models with logit link function were used to assess adjusted odds ratio (aOR) of having a documented screening.

### Results

In total, 126,306 unique patients aged 16 years or older with a total of 291,789 encounters were included. The prevalence of 14-month screening was 79.8% (95% CI, 79.6–80.0). Regression analyses revealed higher screening odds for patients of smaller clinics (<5,000 patients, aOR 1.88; 95% CI 1.80–1.98 vs. clinics >10.000 patients), clinics in areas with mental health provider shortages (aOR 1.69; 95% CI 1.62–1.77), frequent visits (aOR 1.80; 95% CI, 1.78–1.83), and having an annual physical / well child visit encounter (aOR 1.52; 95% CI, 1.47–1.57). Smaller positive effect sizes were also found for male sex, Black or

**Data Availability Statement:** The analysed dataset is not publicly available due to the decision of the

responsible institutional review board. It is however available upon reasonable request within a data sharing agreement. Inquiries can be made to honestbroker@spectrumhealth.org.

**Funding:** Funding/Support: FM received funding as Peter C. and Pat Cook endowed Clinical Research Fellow for conducting this study. OA received funding from the NIH/NCCIH (R00AT009156) and Michigan State University. Role of the Funder/Sponsor: The funding sources had no role in the design and conduct of the study; collection, management, analysis, and interpretation of the data; preparation, review, or approval of the manuscript; and decision to submit the manuscript for publication.

**Competing interests:** The authors have declared that no competing interests exist.

African American race, Asian race, Latinx ethnicity (ref. White/Caucasians), and having insurance through Medicaid (ref. other private insurance).

## Discussion

The 14-month mental health screening rates have been shown to be significantly lower among patients with infrequent visits seeking care in larger clinics and available mental health resources in the community. Introducing and incentivizing mandatory mental health screening protocols in annual well visits, are viable options to increase screening rates.

## Introduction

Depression is the leading cause of disability and is one of the most common mental illnesses in the United States [1]. While many people with behavioral health concerns have encounters with their primary care providers [2], healthcare professionals in primary care not using validated screening instruments will only identify approximately half of their depressive patients [3, 4] Assessing depression using standardized questionnaire instruments in primary care has been proven effective in initiating mental health care [5]. There is a growing body of evidence indicating that mental health screening decreases clinical morbidity in individuals with depression due to early treatment in response to positive screening results [6]. Early diagnosis and management additionally improve patients' quality of life, decrease healthcare costs, and reduce exacerbations of co-morbid medical conditions [7–9]. The United States Preventative Service Task Force (USPSTF) recommended screening for depression in primary care settings for adults in 2016 [10]. A recently released update now also recommends screening for clinically relevant anxiety symptoms [11]. A large study showed increased depression screening rates and increased rates of depression diagnosis and treatment after the introduction of a screening regimen [12]. Particular racial and ethnic minorities and uninsured patients tend to benefit from rigorous mental health screening and subsequent treatment [13]. Although the USPSTF recommendation has been in place for 7 years, many primary care providers still perform mental health screening on a "case-by-case" basis rather than consistently [14], and screening rates have been shown to be 59% [12], 67.2% [15], and 88.8% [16] in various clinical settings.

While there is a growing body of evidence highlighting the benefits of regular mental health screening in primary care, studies of sociodemographic factors associated with lower odds of screening have mostly been conducted before [17, 18] or a few years after the release of the first USPSTF recommendations in 2016 [12, 15, 16]. Newer large-scale data on mental health screening prevalence, which reflects the mental health burden associated with the COVID-19 pandemic [19] is lacking. Identifying these factors is important to inform clinical interventions to improve screening and potential diagnosis. This study addresses this in a large cohort of patients.

## Methods

This is a retrospective cohort study of 18 primary care clinics in the Corewell Health healthcare system in West Michigan. This study received approval from the institutional review board of Corewell Health and was deemed non-human subject research (Decision #: 2022–342). The report of the study results adheres to the STROBE guidelines for observational studies [20].

## Setting

Corewell Health is a non-profit managed healthcare organization with 22 hospitals and more than 300 outpatient facilities and is the largest health care provider across the state of Michigan. This study is based in the West Michigan area with its 1.4 million residents in the Grand Rapids-Kentwood-Muskegon Combined Statistical Area [21] and also includes clinics located in adjacent suburban or rural areas. The median household income in this metropolitan region is similar to the overall United States ($69,643 vs $69,717), however, the foreign-born population is 5.7%, which is considerably lower than the national average (13.6%). The proportion of White/Caucasians among the residents of the metropolitan area Grand Rapids-Kentwood-Muskegon is similar to the overall US population (77% vs. 75.8% US average), however, with a lower proportion of Hispanic (9% vs. 18.9% US average) and Black or African American residents (7% vs 13.6% US average). According to US Census data, the number of uninsured residents was 5.2% in the West Michigan region, lower than the US average (9.8%).

This study used data from 18 primary care clinics employing primary care providers on a total of 143.9 full time equivalent positions. Using the rural-urban commuting area classification introduced by the US Department of Agriculture [22], 83.3% of clinics are located in metropolitan areas, 11.1% in small towns or rural areas, and one clinic (5.6%) in micropolitan areas.

## Data sample

Data was retrieved through a Corewell Health honest broker who randomly assigned eight primary care clinics to the study. Ten other clinics were purposefully assigned to roughly match key demographics in the region. Data was extracted by the honest broker from electronic medical records (EMR, Epic HYPERSPACE, Epic Systems Corporation, Verona WI, USA). The number of clinics included in the study were defined together with Corewell Health's Sensitive Data Sharing Workgroup to ensure a most likely representative sample while reducing the risk that data extraction would impede the operability of the EMR system.

The data set was created using specific criteria: All patients aged 16 or older who completed at least one office visit encounter between 10/15/2021 to 12/14/2022 were included. Patients were excluded if they were not established patients at each respective clinic as well as pure clinical resource visits (e.g. immunization, pap-smear test). Established patients were all patients enrolled with a primary care provider of the respective clinic and having at least one encounter before 10/15/2021.

Besides aggregated data on patients, a dataset listing encounters of the included patients was also retrieved to assess screening probability for each encounter. To control for the respective clinic in multivariable models, we also excluded patients who had encounters at various primary care clinics.

We did not apply any exclusions based on previous or newly diagnosed conditions. We expected annual well visits to be a good opportunity to screen patients for mental health problems, and due to patient preference (e.g. appointment conflicting with work or holidays), we deliberately chose a somewhat longer study time period of 14 months. Requested data was de-identified by the honest broker before being provided to the researchers.

## Outcome and covariates

The primary outcome was receiving at least one mental health screening (either PHQ-4/PHQ-9/GAD-7) during the study period. The Patient Health Questionnaire (PHQ-4) is a commonly used ultra-brief screening instrument for depression and anxiety in adults. The PHQ-4 was introduced in the healthcare system as standard of care to screen for clinically relevant

depressive and anxiety symptoms in June 2017. This questionnaire consists of 4 questions that can be answered on a 4-point scale. The first two questions cover the two main diagnostic criteria for depressive disorders according to DSM 5-TR [23] and form a subscale for depression (PHQ-2). The last two questions cover the main criteria for generalized anxiety disorder (GAD-2). The PHQ-4 has been studied in various populations showing measurement invariance across different genders, ages, and cross-cultural groups [24] and has a specificity of 94.5% and sensitivity of 51.6% compared to the more comprehensive 53-item Brief Symptom Inventory [25].

Typically, a medical assistant would document PHQ-4 responses as part of rooming the patient before the visit. The EMR system Epic prompts a PHQ-4 for every visit, however providers can skip entering results. Epic provides distinct fields to enter PHQ-4 responses and calculates sum scores automatically. Screened patients with a sum score $\geq$ 6, indicating clinically relevant depressive or anxious symptoms, were additionally asked to complete the more comprehensive PHQ-9 [26] and GAD-7 [27] as part of standard clinic procedure.

Covariates include the sociodemographic factors patient's age, sex, self-reported race and ethnicity, and health insurance type. Patient self-reported race/ethnicity was categorized using a two-step approach. Those identifying as Hispanic were labeled "Latina/Latino/Latinx," while non-Hispanic individuals were classified by their specific race (White/Caucasian, American Indian/Alaska Native, Asian, Black or African American, Other). All sociodemographic data were extracted for patients' first encounter during the study period. Furthermore, illness-related factors such as presence of diabetes mellitus (ICD-10 Codes E10, E11, O24.4), chronic ischemic heart disease (I25), chronic rheumatic/inflammatory disease (D89.9, M30-M36, K50-K52, K75.4), malignancy and cancer diagnosis (C00-D48), and mental diagnoses (F) except F17.2 (tobacco use) during or before the study period were included. Chronic conditions were chosen specifically as they have been linked with higher prevalence of anxiety and depression [28–31]. Encounter-related factors included the type of appointment, differentiating between annual physical appointments (such as well child visits for 16 to 21 year-olds or Medicare Annual Wellness Visits [32]) and regular primary care encounters. To adjust for clinics' characteristics associated with the mental health needs of the served community, we collected information on mental health provider shortage using the Health Professional Shortage Area (Mental Health Area HPSA) score on a municipality/county level. Mental HPSAs scores can range between 0 and 25 and are calculated considering a variety of factors including population-to-provider ratio, proportion of residents below federal poverty level (FPL), proportion of residents aged >65 or <18, alcohol and substance use disorder prevalence as well as expected travel time to Nearest Source of Care (NSC) outside the HPSA area [33]. We collapsed Mental HPSAs scores into three groups indicating no mental health provider shortage (HPSA Score = 0), moderate mental health provider shortage (HPSA Score = 1–12), and significant mental health shortage (HPSA Score = 13–25).

Furthermore, we included clinic size calculated as the total of unique patients served during the study period. Clinic size was grouped into small clinics (<5,000 patients), mid-sized clinics (5,000–10,000 patients), and large clinics (>10,000 patients). As number of served patients at a clinic was positively correlated with urban clinic location (Spearman's rho = 0.39) which may introduce collinearity, we deliberately included only clinic size in our multivariable model.

## Statistical analyses

We used descriptive statistics including absolute and relative frequencies, mean, and standard deviation (SD) to characterize our sample. Documentation prevalence for mental health screening during the 14-month study period is displayed in forest plots for various subgroups with corresponding 95% confidence intervals (95% CI).

Lastly, we performed multivariable analysis using general linear models with a logit link function to assess the association of sociodemographic (age, sex, race, ethnicity, health insurance), encounter related (having an annual physical), location related (clinic size and mental health provider shortage at municipal/county level) as well as condition related (diabetes mellitus, chronic ischemic heart disease, cancer diagnosis, mental health diagnosis) factors and having at least one mental health screening (PHQ-4/GAD-7/PHQ-9) documented in a 14-month period as a binary outcome. We applied two different models, where model 1 included all aforementioned factors and model 2 additionally adjusted for frequencies of encounters. Results are reported as adjusted odds ratios (aOR) with corresponding 95% confidence intervals (95% CI). Subjects with missing data were excluded from all logistic regressions.

In all analyses, p values <0.05 were considered as statistically significant. All statistical analyses were performed using SPSS 28 (IBM, Armonk, NY, USA). Figures were plotted using GraphPad Prism 9.5 (GraphPad Software, San Diego, CA, USA).

## Results

After excluding patients with encounters at multiple primary care clinics and clinical resource encounters, 126,306 patients with 291,789 encounters were included for analysis (Fig 1).

Those patients were between 16 and 104 years old (mean: 50.8, SD 18.9), more often of female sex (57.6%) and predominantly White or Caucasian (85.7%).

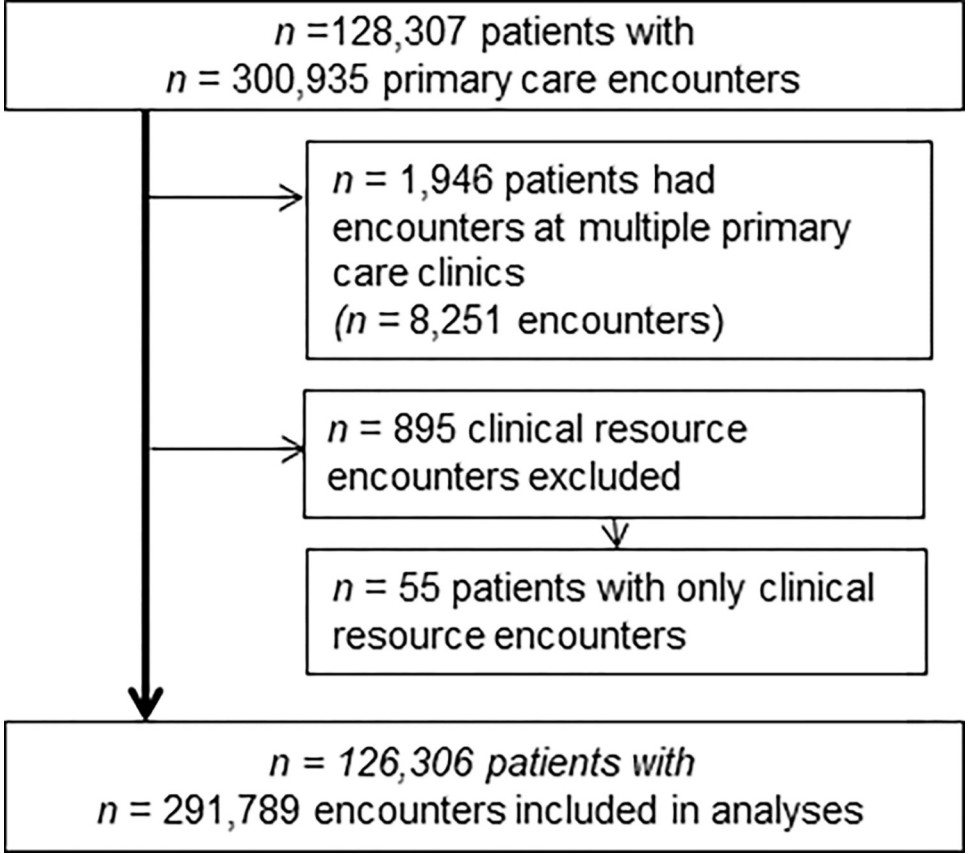

**Fig 1. Flow chart of patient inclusion.**

**Table 1. Sociodemographics of n = 126,306 patients.**

| Characteristic | | No. (%) |
|---|---|---|
| Sex | Male | 53,531 (42.2) |
| | Female | 72,771 (57.6) |
| Age | Mean (SD) | 50.8 (18.9) |
| | 16–35 | 32,731 (25.9) |
| | 36–55 | 37,965 (30.1) |
| | 56–75 | 43,265 (34.3) |
| | 75+ | 12,345 (9.8) |
| Race / Ethnicity[b] | Other | 1,788 (1.4) |
| | American Indian or Alaska Native | 368 (0.3) |
| | Asian | 2,856 (2.3) |
| | Black or African American | 5,461 (4.4) |
| | White or Caucasian | 106,924 (85.7) |
| | Latino/Latina/Latinx | 7,304 (5.9) |
| Insurance | None | 1,580 (1.3) |
| | Medicare | 36,111 (28.6) |
| | Medicaid | 12,079 (9.6) |
| | Private Insurance | 76,536 (60.6) |
| Encounters | Mean (SD) | 2.3 (1.7) |
| | 1–2 encounters | 85,340 (67.6) |
| | 3–4 encounters | 28,796 (22.8) |
| | 5–9 encounters | 11,441 (9.1) |
| | 10+ encounters | 729 (0.6) |
| | Annual physical / Well Child Visit | 86,114 (68.2) |
| Diabetes mellitus | | 17,109 (13.5) |
| Chronic ischemic heart disease | | 7,731 (6.1) |
| Chronic inflammatory / rheumatic condition | | 5,036 (4.0) |
| Cancer diagnosis | | 23,206 (18.4) |
| Mental health diagnosis | | 58.646 (46.4) |
| Clinic size | < 5,000 patients | 20,427 (16.2) |
| | 5,000–10,000 patients | 44,797 (35,5) |
| | > 10,000 patients | 61,082 (48.4) |
| Mental health provider shortage | None (HPSA 0) | 48,553 (38.4) |
| | Moderate (HPSA 1–12) | 42,909 (34.0) |
| | Significant (HPSA 13–25) | 34,844 (27.6) |

[a] Missing sex n = 4

[b] Missing race/ethnicity n = 1,605, "other" includes Native Hawaiian or other Pacific Islander

The primary care clinics were mainly located in metropolitan areas (83.3%) and provided care to an average of 7,017 (min 816, max 23,288, SD 5,613) unique patients during the 14-month study period. Six clinics were situated in counties/municipalities without a mental health provider shortage, while 4 and 8 clinics were located in areas with a moderate or significant mental health provider shortage, respectively. More detailed data are included in Table 1.

On an encounter level, the odds of having any documented PHQ-4/PHQ-9/GAD-7 screening was 66.2% (SD 0.47). On a patient level, 79.8% (95% CI, 79.6–80.0) had a documentation of any PHQ-4/PHQ-9/GAD-7 screening during the 14-month period. 14.3% (n = 14,444) of patients with a documented PHQ-4 screening during the study period were also screened with

a GAD-7. Similarly, 10.6% (n = 10,718) of PHQ-4 screened patients received an additional screening with the PHQ-9.

The patient level 14-month screening prevalence (any PHQ-4/PHQ-9/GAD-7) ranged between 51.3% and 98.6% among the included clinics.

The majority of patients had only one (29.3%) or two (28.1%) encounters. Screening prevalence for patients having a single encounter during the 14-month study period was 67.8% (95% CI, 67.5–68.2), for those with two encounters 83.4 (95% CI, 83.0–83.8).

The 14-months screening prevalence rate was lowest among 34-year-olds (75.7%, n = 1,940) and highest among 87-year-olds (88.4%, n = 466). Screening rates were above average among 16 to 18-year-olds and–with few exceptions–among people aged 60 and older (S1 Fig).

In unadjusted analyses, odds of receiving a screening within 14 months were higher for patients who were males (80.1%, 95% CI 79.8–80.5), had an annual physical appointment (81.2%, 95% CI 81.0–81.5 vs. those who did not 76.6%, 95% CI 76.2–77.1) and Black or African American patients had higher screening rates (86.2%, 95% CI 85.3–87.1) vs. White or Caucasian patients (79.1%; 95% CI 78.9–79.4). Patients with up to two appointments had lower (73.9%; 95% CI 73.6–74.2) than average chances to receive mental health screening. Patients with no chronic conditions (diabetes mellitus, chronic ischemic heart disease, chronic inflammatory condition, cancer diagnosis or mental health problems) had 14-month screening rates below average. More information is provided in Fig 2.

Two different multivariable logistic regression models including a total of n = 124,698 patients were performed. All aORs are depicted in Table 2. Model 2 accounts for frequent encounters, while model 1 does not. Both models revealed sex differences on the outcome "Documented mental health screening (any PHQ-4/GAD-7/PHQ-9) within 14 months" in favor of male patients. Also, Black or African Americans, Asians, and Latinas/Latinos/Latinx showed significantly higher odds to be screened compared to White or Caucasians in both models, respectively. Compared to patients with private insurance, those insured through Medicaid were more likely to have a screening documented. Furthermore, patients having at least one annual physical / wellness visit / well child visit had higher screening odds compared to those who did not have one of these preventive encounters.

Patients' aged 20 to 49 were less likely to have a documented mental health screening in model 1 compared to the reference group of 60-69-year-olds, but when further adjusted to frequency of encounters the effect size decreased and remained only statistically significant for 20-29-year-olds.

The association between clinic size/location and mental health screening prevalence shown in descriptive analyses was revealed in both regression models. Smaller clinics were more likely to perform and document mental health screening compared to larger clinics, and clinics located in areas with a mental health provider shortage performed those screenings significantly more often on their patients. While patients with a mental health diagnosis showed increased screening rates in both regression models compared to those without such a diagnosis, screening prevalence did not differ among patients with diagnosed diabetes or chronic ischemic heart disease, when adjusting for frequent encounters. For cancer patients and patients with chronic inflammatory or rheumatic conditions, screening odds were even lower compared to those without such conditions when adjusted for frequency of encounters.

## Discussion

While early detection of mental health disorders has been shown to enhance patients' quality of life, reduce healthcare costs, and subsequently decrease complications from co-morbidities,

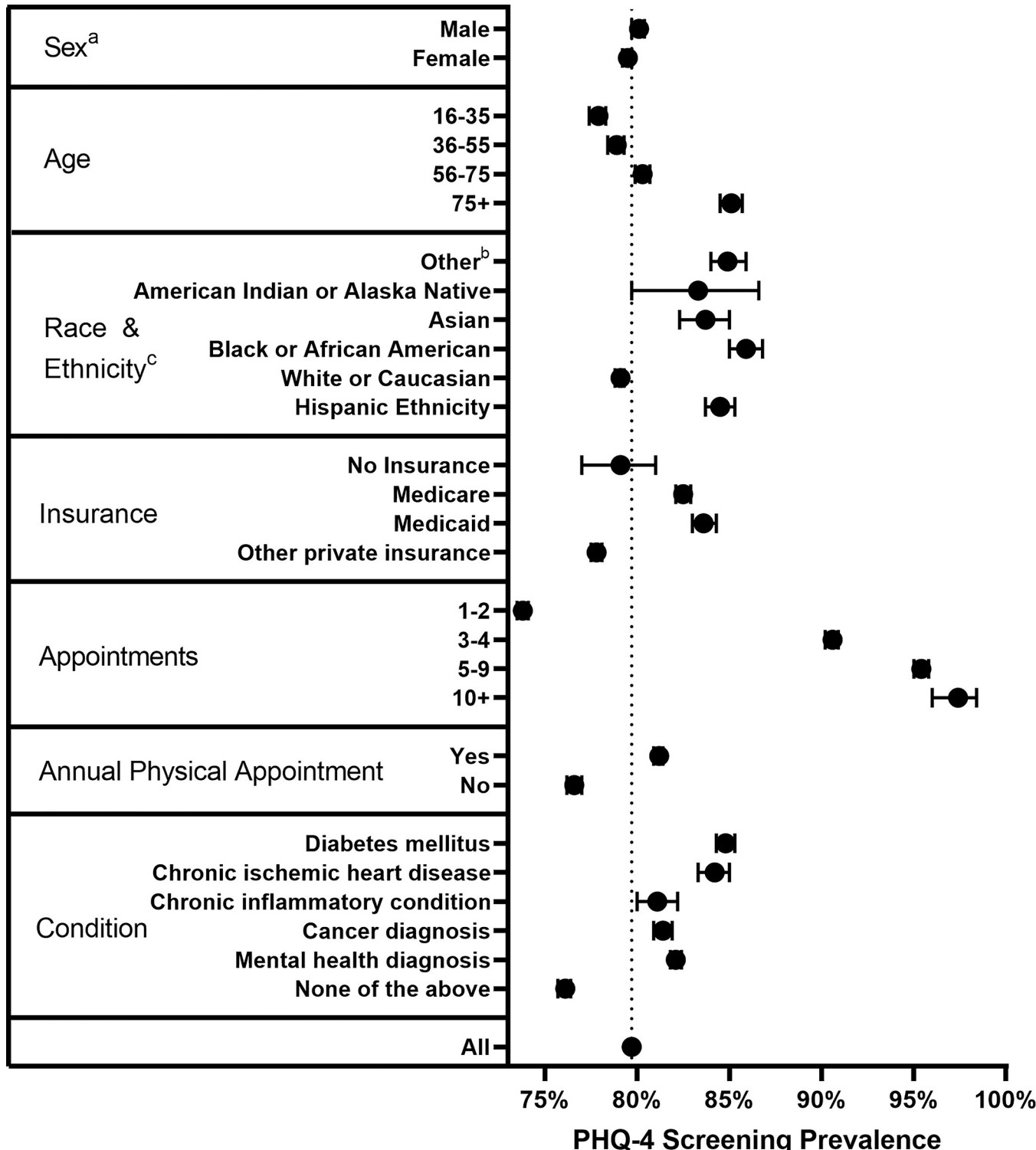

**Fig 2. Unadjusted 14-months mental health screening prevalence with corresponding 95% CI.** [a] Missing sex n = 4, [b] Includes Native Hawaiian or other Pacific Islander, [c] Missing race / ethnicity n = 1,605.

Table 2. **Results of binary logistic regression models (n = 124,698 patients).**

| Parameter | | Model 1 | | Model 2 | |
|---|---|---|---|---|---|
| | | aOR (95% CI) | P | aOR (95% CI) | P |
| Age (years) | 16–19 | 1.11 (1.02–1.21) | 0.019 | 1.24 (1.13–1.35) | <0.001 |
| | 20–29 | 0.84 (0.79–0.89) | <0.001 | 0.94 (0.89–1.00) | 0.050 |
| | 30–39 | 0.86 (0.81–0.90) | <0.001 | 0.95 (0.89–1.00) | 0.053 |
| | 40–49 | 0.95 (0.89–1.00) | 0.048 | 1.01 (0.96–1.07) | 0.694 |
| | 50–59 | 0.96 (0.91–1.01) | 0.087 | 0.97 (0.92–1.03) | 0.307 |
| | 60–69 | ref | | ref | |
| | 70–79 | 1.08 (1.01–1.14) | 0.016 | 1.03 (0.96–1.09) | 0.404 |
| | 80–89 | 1.60 (1.47–1.75) | <0.001 | 1.37 (1.25–1.50) | <0.001 |
| | 90+ | 1.42 (1.20–1.67) | <0.001 | 1.26 (1.07–1.49) | 0.007 |
| Sex | Male | 1.10 (1.06–1.13) | <0.001 | 1.14 (1.11–1.18) | <0.001 |
| | Female | ref | | ref | |
| Race/Ethnicity | Latina/Latino/Latinx | 1.74 (1.62–1.86) | <0.001 | 1.61 (1.50–1.73) | <0.001 |
| | Other[a] | 1.15 (1.01–1.30) | 0.029 | 1.09 (0.96–1.23) | 0.198 |
| | American Indian or Alaska Native | 1.25 (0.94–1.64) | 0.120 | 1.18 (0.89–1.57) | 0.253 |
| | Asian | 1.38 (1.24–1.53) | <0.001 | 1.33 (1.20–1.48) | <0.001 |
| | Black or African American | 1.37 (1.27–1.49) | <0.001 | 1.20 (1.10–1.31) | <0.001 |
| | White or Caucasian | ref | | ref | |
| Insurance | None | 1.08 (0.95–1.22) | 0.255 | 1.14 (1.00–1.29) | 0.052 |
| | Medicare | 1.13 (1.07–1.19) | <0.001 | 1.03 (0.97–1.09) | 0.345 |
| | Medicaid | 1.31 (1.24–1.38) | <0.001 | 1.16 (1.09–1.22) | <0.001 |
| | Other private insurance | ref | | ref | |
| Annual Physical / Well Child Visit | | 1.49 (1.44–1.53) | <0.001 | 1.35 (1.31–1.39) | <0.001 |
| Encounters | (continous) | | 1.80 (1.78–1.83) | <0.001 | |
| Diabetes mellitus | | 1.37 (1.31–1.44) | <0.001 | 1.00 (0.96–1.06) | 0.861 |
| Chronic ischemic heart disease | | 1.06 (0.99–1.14) | 0.093 | 0.94 (0.87–1.00) | 0.066 |
| Chronic inflammatory / rheumatic diagnosis | | 1.02 (0.94–1.09) | 0.694 | 0.89 (0.82–0.96) | 0.002 |
| Malignancy / Cancer diagnosis | | 1.01 (0.97–1.05) | 0.527 | 0.93 (0.89–0.97) | <0.001 |
| Mental health diagnosis | | 1.37 (1.33–1.42) | <0.001 | 1.12 (1.08–1.15) | <0.001 |
| Clinic size (unique patients/14 months) | <5,000 patients | 1.95 (1.87–2.05) | <0.001 | 1.88 (1.80–1.98) | <0.001 |
| | 5,000–10,000 patients | 1.58 (1.51–1.65) | <0.001 | 1.62 (1.55–1.69) | <0.001 |
| | >10,000 patients | ref | | ref | |
| Mental Health Provider Shortage (HPSA) | None | ref | | ref | |
| | Moderate | 1.95 (1.88–2.02) | <0.001 | 1.96 (1.89–2.03) | <0.001 |
| | Significant | 1.59 (1.52–1.66) | <0.001 | 1.69 (1.62–1.77) | <0.001 |

[a] "Other" includes Native Hawaiian or other Pacific Islander

our study suggests that providers struggle to screen patients with only occasional encounters, younger patients in their 20s, female sex, White and Caucasian race, having private insurance, and not having preventive annual physical visits. This is concerning as the highest prevalence rates for depression are found among people in their 20s, with women particularly affected at this age [34].

While 14-month screening prevalence in our study was 79.8%, other researchers found considerably lower depression screening rates of 48.6% using a representative sample of US adults aged 35 or older [18]. That study, however, used patient self-reported data and found other demographic factors correlated with mental health screening than in our study. Another

study reported similar mental health screening rates, with higher odds in patients aged 65 to 84 and 85+, compared to those aged 18 to 44, however, insurance type was not considered in that study [35].

Our study revealed significant screening rate variations among clinics, despite the health system having a longstanding standard procedure for mental health screening. This heterogeneity in one geographically bounded health system is surprising and concerning. Implementing mental health screening in the work routine was considerably more successful in some clinics than in others. This raises not only questions on structural and organizational barriers that may contribute to under-screening in those clinics but also indicates that a single top-down implementation effort may not be effective among some clinics which may require some more tailored and continuous approaches, e.g. constant monitoring and feedback, incentivizing of screening, highlighting the importance of screening in team meetings, education of office staff, reminder through EMR software, etc. To our surprise, smaller clinics and clinics in areas with mental health provider shortages had substantially higher mental health screening odds. While this phenomenon has to our knowledge not been described before, it is likely that providers in communities lacking mental health providers and with fewer options to refer patients with mental health needs may be more aware of their patients' mental health problems and perhaps even more keen in treating them. Conversely, providers in areas with sufficient mental health services may be more prone to expect their patients with mental health problems to seek treatment by specialists and thus do not prioritize mental health screening. While scarcity of mental health services is expanding in the U.S. [36], further qualitative research can provide insights on how primary care providers in both provider shortage areas and sufficiently covered areas perceive their professional roles when treating mental health problems. Prior research indicated barriers to primary care providers to screen for mental health, such as lack of training on mental health care [37], anticipated negative reactions to screening from patients, perception of "knowing a patient personally" [38], time constraints [14] and low comfort levels of primary care providers' comfort in treating psychiatric conditions [39].

Higher screening rates in the elderly may be attributed to Medicare Annual Wellness Visits (AWV) introduced in 2011, offering preventive services that include mental health screening without patient co-payments [32]. The established practice of well child visits, which are regularly offered until the age of 21, and also include screening for mental health, could have also a positive effect on screening prevalence [40]. Unlike AWVs, preventive encounters (e.g. routine annual physicals) in private health insurance plans may either not mandate mental health screenings or propose it but not enforce actual implementation, which may explain lower screening odds.

Furthermore, providers can be prone to bias about who they think may "need" screening, and providers may not be aware of racial and cultural differences in the symptom presentation [41]. However, our study found significantly higher screening rates among Black and African American, Asian and Latinx patients compared to Whites and Caucasian. While this may reflect increased efforts of diversity equity and inclusion initiatives to address mental health disparities in minority populations, other researchers highlighted that implicit bias is a possible explanation that Black people have higher rates of being diagnosed with certain mental diagnoses [42]. Other studies also highlighted lower screening rates among some limited English language proficiency patient groups [16, 43].

Patients with chronic conditions that are associated with higher mental health burdens [28–30] did have higher mental health screening prevalence in unadjusted analyses, but average or even lower odds of receiving mental health screening when adjusted for frequent visits suggesting that encounter frequency is a modifying factor. While our study could show this tendency for various conditions, patients with a known mental health diagnosis were more likely to receive a screening in both regression models.

Quality improvement projects introducing standard of care procedures, treatment algo-rithms, and/or modified office protocols have shown to increase mental health screening [16, 35, 44]. While these efforts remain on a local level, our study suggested that annual wellness visits correlate with increased screening rates. Encouraging and incentivizing wellness visits, along with incorporating mental health screening as a standard of care procedure, could have a nationwide impact. Timely diagnosis and treatment of depression and anxiety symptoms are beneficial in terms of prognostic outcomes, can prevent worsening of comorbidities, and have been proven cost-effective [7, 8, 45].

## Limitations

Our study comes with limitations that need to be considered when interpreting the results. The clinics chosen were not similar to the sociodemographic structure of the overall U.S. population, for example, our patient sample had a higher proportion of White or Caucasian patients. This limits the validity for certain racial and ethnic minorities. Another limitation is that our study only encompassed clinics from one health care system in the Midwest. Further-more, the data did not specify if patients rejected mental health screening or were not able to be screened, e.g. due to disabilities.

## Conclusions

This study highlights clinic size, mental health service availability in the community, and num-ber of encounters of a patient are main determinants for receiving the recommended mental health screening. Primary care providers offer accessible mental health services by assessing and diagnosing problems, provide treatment or referrals if necessary, and support patients in managing their condition. As a prerequisite, providers need to be well trained, aware of their important role and comfortable in treating patients. Mental health screening was performed more often by providers of smaller clinics with scarce mental health treatment resources in their communities, suggesting that these providers were more likely to fulfil this role. On the contrary, healthy and younger patients with less healthcare encounters seeking care in larger urban clinics in areas with generally good availability of mental health services were less likely to receive screening for anxiety and depression. While other factors like sex, race or ethnicity also play a role, this finding is particularly concerning given the high rates of depression and anxiety in these age groups.

Furthermore, our study showed a considerable variety of mental health screening preva-lence between clinics. This highlights the necessity for a continuous and tailored implementa-tion effort to foster equitable mental health care.

## Supporting information

**S1 Fig. 14-month documented mental health screening prevalence with corresponding 95% CI among different ages (patients aged >95 years were grouped into 95-year-olds).** Dotted line depicts mean screening rate.
(DOCX)

## Author Contributions

**Conceptualization:** Judith E. Arnetz, Omayma Alshaarawy, Harland T. Holman.

**Data curation:** Frank Müller.

**Formal analysis:** Frank Müller.

**Investigation:** Frank Müller, Alyssa M. Abdelnour, Diana N. Rutaremara.

**Methodology:** Frank Müller, Omayma Alshaarawy.

**Project administration:** Frank Müller.

**Supervision:** Judith E. Arnetz, Eric D. Achtyes, Harland T. Holman.

**Visualization:** Frank Müller.

**Writing – original draft:** Frank Müller, Alyssa M. Abdelnour, Diana N. Rutaremara, Omayma Alshaarawy, Harland T. Holman.

**Writing – review & editing:** Frank Müller, Alyssa M. Abdelnour, Diana N. Rutaremara, Judith E. Arnetz, Eric D. Achtyes, Omayma Alshaarawy, Harland T. Holman.

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
