## [Decision Letter · Decision Letter 0]

20 Nov 2023

PONE-D-23-16932The Impact of Sociodemographic Factors on Mental Health Screening Rates in Primary CarePLOS ONE

Dear Dr. Müller,

Thank you for submitting your manuscript to PLOS ONE. After careful consideration, we feel that it has merit but does not fully meet PLOS ONE’s publication criteria as it currently stands. Therefore, we invite you to submit a revised version of the manuscript that addresses the points raised during the review process.

We look forward to receiving your revised manuscript.

Kind regards,

Dr, Swaantje Casjens

Academic Editor

PLOS ONE

Journal Requirements:

3. Please expand the acronym “NIH/NCCIH” (as indicated in your financial disclosure) so that it states the name of your funders in full.

Additional Editor Comments:

Please note in particular the comments on the statistical analysis from both reviewers: modelling of age, clinic-related heterogeneity. I would also like to point out that a p-value <0.05 represents statistical significance and not general significance.

Also make sure that all abbreviations are introduced the first time they are used (e.g., EMR in the abstract) and check the references carefully. It does not conform to PlosOne specifications, e.g., journal abbreviations are required.

Reviewers' comments:

Reviewer's Responses to Questions

**Comments to the Author**

1. Is the manuscript technically sound, and do the data support the conclusions?

Reviewer #1: Partly

Reviewer #2: Partly

2. Has the statistical analysis been performed appropriately and rigorously? 

Reviewer #1: No

Reviewer #2: Yes

3. Have the authors made all data underlying the findings in their manuscript fully available?

Reviewer #1: No

Reviewer #2: No

4. Is the manuscript presented in an intelligible fashion and written in standard English?

Reviewer #1: Yes

Reviewer #2: Yes

5. Review Comments to the Author

Reviewer #1: Overall I found this manuscript to be an important addition to the literature on mental health screening. The questions asked are sound and the writing is generally clear. Specific attention should be paid to the analysis of age in the model and the interpretation of the chronic disease data. Detailed comments on both of these along with other minor comments below:

Abstract

In the design section of the abstract the outcome is listed as “documentation of a PHQ-4 screening” and the analysis strategy is stated as “assess odds ratio (aOR) of receiving a screening.” I would recommend changing tis to “assess odds ration (aOR) of having a documented screening” to be consistent.

Also please clarify in the abstract that outcome is the documentation of screening at the patient (rather than visit) level.

Discussion of abstract has the take home point that “Introducing and incentivizing mandatory mental health screening protocols in annual well visits, similar to well child visits or Medicare Annual Wellness Visits, are viable options to increase screening rates.” I am confused by this statement because the the design section states that mental health screening has been a standard procedure in this health system since prior to the start of this study period. These screenings are already incentivized by most insurers.

Introduction

Paragraph one contains the sentence “A large study showed increased depression screening rates and increased rates of depression diagnosis and treatment after the introduction of a screening regime.” I believe the last word of this sentence should be “regimen” rather than “regime.”

Setting

Suggest removing the word “somewhat” from the following sentence: “with a somewhat lower proportion of Hispanic (9% vs. 18.9% US average) and Black or African American residents (7% vs 13.6% US average).” These are both roughly half the proportion which I wouldn’t classify as somewhat.

Data sample

Please define “established patient.”

Outcome and covariates. Last paragraph starts with “Covariates include the sociodemographic factors patient’s age (at first encounter)”. Can delete “(at first encounter)” since you specify in the next sentence that this is true for all covariates.

Results

I am a little confused by the reporting of patient level outcomes and encounter level outcomes. For example on page 8, second paragraph, does the following sentence refer to the prevalence of screening at the encounter level or patient level? “The 14-month screening prevalence ranged between 51.3% and 98.6% among the included clinics.” Can the authors clarify this by adding the words “at the patient level” or “at the encounter level” after the word prevalence?

First sentence of paragraph 3 on page 8 should end with the word “encounters” rather than “encounter.”

I question the authors’ choice to include age as a continuous variable in the model when the unadjusted analysis shows a bimodal distribution (higher screening rates at younger and older ages). I suggest that they work with a statistician to find a more appropriate way to treat age in the multivariate analysis, perhaps categorically as it is presented in Table 1. I think this is especially important given that age does seem to be an important factor in screening rates and that modelling this continuously shows, I believe misleadingly, that it is not (unadjusted model shows significance but with an odds ratio of essentially 1).

Interesting that for all chronic diseases the odds of receiving a PHQ4 decreases when you adjust for frequency of visits, suggesting that the effect is indirect (chronic disease leads to increased number of visits which leads to increased screening rates) rather than direct (chronic disease leads to increased screening). This seems to be the main take home point from the chronic disease analysis and I don’t see it discussed in the paper.

Discussion

Again, I think that the conclusions drawn from the age analysis are misleading based on the treatment of this covariate as a linear variable in the model. This will be important to address given the author’s discussion of the interaction between age and wellness visits in the conclusion (which I think is an interesting point that should be explored through proper analysis of the bimodal age data).

Again, conclusions from chronic disease analysis seem, to me, to be that frequency of visits is a modifying factor (all in the same direction, increased screening with increased frequency of visits) and that, when this is adjusted for, different chronic diseases have different associations with screening rates (none for diabetes and CV disease, decreased rates of screening for inflammatory disease and cancer, and increased screening for mental health).

Figures

It is somewhat disorienting to me that Figure 2 has PHQ4 prevalence on the y axis and Figure 3 has PHQ prevalence on the x axis. Can the authors be consistent in the presentation of their data? Figure 2 seems overly detailed to me given the re-presentation of this data in Figure 3 with categories that tell the same story.

Reviewer #2: This is an interesting article reporting on the important matter of screening for depression and anxiety in primary care settings.

It conducts a retrospective analysis of EMR data from a health system in Western Michigan, aiming to shed light on some of the associations between patient characteristics and their receipt of the PHQ-4 screening questionnaire.

As you will see in my comments, I believe that a set of additional analyses could really elevate this paper and turn it into an important contribution. Mainly, this has to do with the need to separate the associations found between patients’ demographics and their receipt of the screening questionnaire, and the very substantial heterogeneity detected in the rate of questionnaire administration between different clinics (an important finding on its own).

The question whether, upon revision, the authors can disentangle these drivers of the observed association will make or break this article.

Comments for the authors:

1. On p. 8, the authors report that “The 14- month screening prevalence ranged between 51.3% and 98.6% among the included clinics.” This indicates huge variance and should be treated as a key finding on its own that raises important questions concerning the differences between these clinics. The authors need to use their data to investigate this heterogeneity as it might in fact drive the demographic effects reported. What are the sources of this heterogeneity? Does the data allow you to identify individual providers and their likelihoods of administering the questionnaire? The analyses looking at patient characteristics should additionally control for clinic-related covariates (for one, the authors mention that they tag clinics by rurality). It is important to disaggregate observed effects between those having to do with patient characteristics and those driven by place. Additionally, on p. 7 the authors mention a sensitivity analysis that looked at between-clinic differences, but its findings are not reported nor discussed. The clinic-driven heterogeneity of results is germane and should be addressed both descriptively and in the discussion.

2. Adding to the same between-clinic heterogeneity issue, on p. 5 the authors write that “To control for the respective clinic in multivariable models, we also excluded patients who had encounters at various primary care clinics.” I am not sure I understand the logic for excluding these clients. In fact, an analysis that focuses on these patients in particular might be useful to uncover the puzzle of the between-clinic gaps.

3. On p. 6, authors write that “The EMR system Epic prompts a PHQ-4 for every visit, however providers can skip entering results. Epic provides distinct fields to enter PHQ-4 responses and calculates sum scores automatically. Screened patients with a sum score ≥ 6, indicating clinically relevant depressive or anxious symptoms, were additionally asked to complete the more comprehensive PHQ-9 and GAD-7 as part of standard clinic procedure.”

Instead of the PHQ-4, do any patients receive these more comprehensive questionnaires or undergo some other mental health screening during their visits? If so, the outcome variable and analyses need to be amended.

Additionally, of the patients who received the PHQ-4, examining how many scored 6 or over (or, if the scores are inaccessible, how many were given these more comprehensive questionnaires following the PHQ-4) could be an interesting outcome further elucidating the likelihood that different demographics’ mental illness goes undiagnosed.

4. Throughout the article, the use of the term “impact” (e.g. “The Impact of Sociodemographic Factors on Mental Health Screening”) should be changed to “association”. The evidence provided does not establish causality.

5. Given the centrality of race to the framing of the article, I would include the Asian patient odds in the text, in addition to other racial groups (e.g., Abstract and p. 8). I believe the rates for women patients, too, are missing from p. 8.

6. The discussion is currently too long. Precious word-count can be diverted to conducting and reporting further analyses, as suggested above. That being said, I am convinced by the point raised in the discussion that “limited mental health treatment capacities might reduce providers’ willingness to screen for mental health problems as they find themselves in a position where they cannot refer patients to specialists” (p. 11). I think this argument as well as the phenomenon of psychiatrist scarcity indicated by reference 42 could certainly be one of the mechanisms driving the between-clinic disparities this study observes. If indeed more rural clinics have lower rates of questionnaire administration, this would support this theory.

7. On p. 5, the authors mention that “Patients were excluded if they were not established patients at each respective clinic”. Why were these patients excluded? Was it necessary? I don’t believe Figure 1 properly reflects this exclusion.

More minor comments:

1. On p. 3, the sentence starting with “While there is a growing body of evidence” is missing some words.

2. On p. 8, authors report that “The 14-months screening prevalence rate was lowest among 34-year-olds (75.4%, n=1,940) and highest among 87-year-olds (88.4%, n=466).” It might be preferable to report (and statistically test) differences in questionnaire administration frequency on wider, 5- or 10-year age spans.

3. In the Discussion, on p. 10, the authors mention that “Our study showed that within a 14-month period, 79.7% of patients received a PHQ-4 screening with considerably lower rates for patients in their 20’s and 30’s.” This should be statistically tested.

I thank you for sharing this manuscript with me and wish you good luck in continuing your analysis of this interesting data.

6. PLOS authors have the option to publish the peer review history of their article (what does this mean?). If published, this will include your full peer review and any attached files.

Reviewer #1: No

Reviewer #2: No

---

## [Author Response · Author response to Decision Letter 0]

20 Jan 2024

Reviewer 1

Overall I found this manuscript to be an important addition to the literature on mental health screening. The questions asked are sound and the writing is generally clear. Specific attention should be paid to the analysis of age in the model and the interpretation of the chronic disease data. Detailed comments on both of these along with other minor comments below:

Thank you for your time and efforts in reviewing our paper.

Abstract

In the design section of the abstract the outcome is listed as “documentation of a PHQ-4 screening” and the analysis strategy is stated as “assess odds ratio (aOR) of receiving a screening.” I would recommend changing tis to “assess odds ration (aOR) of having a documented screening” to be consistent.

We agree and changed the wording.

Also please clarify in the abstract that outcome is the documentation of screening at the patient (rather than visit) level.

Thank you for this remark. We have added this important aspect.

Discussion of abstract has the take home point that “Introducing and incentivizing mandatory mental health screening protocols in annual well visits, similar to well child visits or Medicare Annual Wellness Visits, are viable options to increase screening rates.” I am confused by this statement because the the design section states that mental health screening has been a standard procedure in this health system since prior to the start of this study period. These screenings are already incentivized by most insurers.

Thank you for this remark. While some insurers include mental health screening in their wellness visits, not all do so. Furthermore, private insurance providers typically don't invest considerable efforts in ensuring or enforcing the actual implementation of mental health screening, despite proposing its inclusion in coverage. In addition, healthcare systems may establish standard care procedures, yet clinics might deprioritize these protocols, particularly when there are no sanctions for non-compliance. Conversely, Annual Wellness Visits (AWVs) offer a high revenue value unit rate, providing a strong incentive for conducting and documenting mental health screening as an integral part of the AWV process, which is mandatory for billing. We have added this to the discussion:

“Higher screening rates in the elderly may be attributed to Medicare Annual Wellness Visits (AWV) introduced in 2011, offering preventive services that include mental health screening without patient co-payments.32 The established practice of well child visits, which are regularly offered until the age of 21, and also include screening for mental health, could have also a positive effect on screening prevalence.38 Unlike AWVs, preventive encounters (e.g. routine annual physicals) in private health insurance plans may either not mandate mental health screenings or propose it but not enforce actual implementation, which may explain lower screening odds.”

Introduction

Paragraph one contains the sentence “A large study showed increased depression screening rates and increased rates of depression diagnosis and treatment after the introduction of a screening regime.” I believe the last word of this sentence should be “regimen” rather than “regime.”

Thank you for catching this typo. Corrected.

Setting

Suggest removing the word “somewhat” from the following sentence: “with a somewhat lower proportion of Hispanic (9% vs. 18.9% US average) and Black or African American residents (7% vs 13.6% US average).” These are both roughly half the proportion which I wouldn’t classify as somewhat.

We agree and have deleted “somewhat”.

Data sample

Please define “established patient.”

Thank you for catching this. We have added this information to the Methods section:

“Established patients were all patients enrolled with a primary care provider of the respective clinic and having at least one encounter before 10/15/2021.”

Outcome and covariates. Last paragraph starts with “Covariates include the sociodemographic factors patient’s age (at first encounter)”. Can delete “(at first encounter)” since you specify in the next sentence that this is true for all covariates.

Done.

Results

I am a little confused by the reporting of patient level outcomes and encounter level outcomes. For example on page 8, second paragraph, does the following sentence refer to the prevalence of screening at the encounter level or patient level? “The 14-month screening prevalence ranged between 51.3% and 98.6% among the included clinics.” Can the authors clarify this by adding the words “at the patient level” or “at the encounter level” after the word prevalence?

We agree that this is indeed confusing. We have changed the sentences accordingly.

First sentence of paragraph 3 on page 8 should end with the word “encounters” rather than “encounter.”

Thank you for catching this error. Corrected.

I question the authors’ choice to include age as a continuous variable in the model when the unadjusted analysis shows a bimodal distribution (higher screening rates at younger and older ages). I suggest that they work with a statistician to find a more appropriate way to treat age in the multivariate analysis, perhaps categorically as it is presented in Table 1. I think this is especially important given that age does seem to be an important factor in screening rates and that modelling this continuously shows, I believe misleadingly, that it is not (unadjusted model shows significance but with an odds ratio of essentially 1).

Thank you for this comment. We agree and have added the patient’s age now as a categorical variable to the multivariable model and re-done the analyses. This revealed the bimodal distribution with lower screening odds in patients 20s (both models) and 20s-40s (without adjusting on frequent encounters) compared to reference (60-69-year-olds). Patients aged 80 and above had significantly higher screening odds. We also changed the discussion accordingly.

Of note, we streamlined the race/ethnicity categorization into a single variable. This adjustment was made because Latinx patients frequently left the "race" field blank, resulting in a decrease in the inclusion of Latinx individuals to the multivariable model due to missing values. We have added this to the methods section.

Furthermore, we have added clinic characteristics (size and location) to the regression model as suggested by reviewer #2, which provided further insights.

Interesting that for all chronic diseases the odds of receiving a PHQ4 decreases when you adjust for frequency of visits, suggesting that the effect is indirect (chronic disease leads to increased number of visits which leads to increased screening rates) rather than direct (chronic disease leads to increased screening). This seems to be the main take home point from the chronic disease analysis and I don’t see it discussed in the paper.

Thank you for highlighting this important aspect. We have added it to the discussion:

“Patients with chronic conditions that are associated with higher mental health burden did have higher mental health screening prevalence in unadjusted analyses, but average or even lower odds of receiving mental health screening when adjusted for frequent visits suggesting that encounter frequency is a modifying factor. While our study could show this tendency for various conditions, patients with a known mental health diagnosis were more likely to receive a screening in both regression models.”

Discussion

Again, I think that the conclusions drawn from the age analysis are misleading based on the treatment of this covariate as a linear variable in the model. This will be important to address given the author’s discussion of the interaction between age and wellness visits in the conclusion (which I think is an interesting point that should be explored through proper analysis of the bimodal age data).

Thank you for this comment, as described above, we have changed the regression model accordingly and now report on the updated results and also discuss new findings. 

Again, conclusions from chronic disease analysis seem, to me, to be that frequency of visits is a modifying factor (all in the same direction, increased screening with increased frequency of visits) and that, when this is adjusted for, different chronic diseases have different associations with screening rates (none for diabetes and CV disease, decreased rates of screening for inflammatory disease and cancer, and increased screening for mental health).

Thank you for this remark. See response above.

Figures

It is somewhat disorienting to me that Figure 2 has PHQ4 prevalence on the y axis and Figure 3 has PHQ prevalence on the x axis. Can the authors be consistent in the presentation of their data? Figure 2 seems overly detailed to me given the re-presentation of this data in Figure 3 with categories that tell the same story.

Thank you for this comment. We believe that figure 2 provides important and detailed information and allows bivariate comparisons, which is of importance to reviewer #2. To keep the presented information concise, we decided however to move figure 2 to the appendix.

 

Reviewer #2

This is an interesting article reporting on the important matter of screening for depression and anxiety in primary care settings.

It conducts a retrospective analysis of EMR data from a health system in Western Michigan, aiming to shed light on some of the associations between patient characteristics and their receipt of the PHQ-4 screening questionnaire.

As you will see in my comments, I believe that a set of additional analyses could really elevate this paper and turn it into an important contribution. Mainly, this has to do with the need to separate the associations found between patients’ demographics and their receipt of the screening questionnaire, and the very substantial heterogeneity detected in the rate of questionnaire administration between different clinics (an important finding on its own).

Thank you for your time and effort in reviewing our manuscript.

The question whether, upon revision, the authors can disentangle these drivers of the observed association will make or break this article.

Comments for the authors:

1. On p. 8, the authors report that “The 14- month screening prevalence ranged between 51.3% and 98.6% among the included clinics.” This indicates huge variance and should be treated as a key finding on its own that raises important questions concerning the differences between these clinics. The authors need to use their data to investigate this heterogeneity as it might in fact drive the demographic effects reported. What are the sources of this heterogeneity? Does the data allow you to identify individual providers and their likelihoods of administering the questionnaire? The analyses looking at patient characteristics should additionally control for clinic-related covariates (for one, the authors mention that they tag clinics by rurality). It is important to disaggregate observed effects between those having to do with patient characteristics and those driven by place. Additionally, on p. 7 the authors mention a sensitivity analysis that looked at between-clinic differences, but its findings are not reported nor discussed. The clinic-driven heterogeneity of results is germane and should be addressed both descriptively and in the discussion.

Thank you for this important comment. We agree that including respective clinics as covariates in our multivariable analyses does not add any additional explanatory value to the model given the considerable variance of screening rates among the clinics. Analyzing screening rates on provider level can be also misleading, as patient care involves a clinical team, with medical assistants often performing PHQ screenings and other preparatory measures such as vital signs, height/weight during rooming.

We have decided to add two important measures to the analyses, that should allow us to better characterize the involved clinics.

a) We have added metrics on Health Professional Shortage Area (HPSA) to include mental health shortages in the respective county/municipality.

b) Furthermore, we added clinic size measured as they served unique patients during the study period. Of note, clinic size was correlated with the RUCA rurality codes thus we dropped RUCA from analyses to avoid factor collinearity.

We have added this to the methods section:

“To adjust for clinics’ characteristics associated with the mental health needs of the served community, we collected information on mental health provider shortage using the Health Professional Shortage Area (Mental Health Area HPSA) score on municipality/county level. Mental HPSAs scores can range between 0 and 25 and are calculated considering a variety of factors including population-to-provider ratio, proportion of residents below federal poverty level (FPL), proportion of residents aged >65 or <18, alcohol and substance use disorder prevalence as well as expected travel time to Nearest Source of Care (NSC) outside the HPSA area.33 We collapsed Mental HPSAs scores into three groups indicating no mental health provider shortage (HPSA Score = 0), moderate mental health provider shortage (HPSA Score = 1-12), and significant mental health provider shortage (HPSA Score = 13-25).

Furthermore, we included clinic size calculated as the total of unique patients served during the study period. Clinic size was grouped into small clinics (<5,000 patients), mid-sized clinics (5,000-10,000 patients), and large clinics (>10,000 patients). As the number of served patients at a clinic was positively correlated with urban clinic location (Spearman’s rho = 0.39) which may introduce collinearity, we deliberately included only clinic size in our multivariable model.”

Furthermore, we updated table 1 and now report also on the characteristics of the clinics in both text and table:

“Six clinics were situated in counties/municipalities without a mental health provider shortage, while 4 and 8 clinics were located in areas with a moderate or significant mental health provider shortage, respectively.”

“The association between clinic size/location and mental health screening prevalence shown in descriptive analyses was revealed in both regression models. Smaller clinics were more likely to perform and document mental health screening compared to larger clinics, and clinics located in areas with mental health provider shortage performed those screenings significantly more often on their patients.”

Furthermore, we have restructured the entire discussion section, now highlighting the new findings as follows:

“Our study revealed significant screening rate variations among clinics, despite the health system having a longstanding standard procedure for mental health screening. To our surprise, smaller clinics and clinics in areas with mental health provider shortages had substantially higher mental health screening odds. While this phenomenon has to our knowledge not been described before, it is likely that providers in communities lacking mental health providers and with fewer options to refer patients with mental health needs may be more aware of their patients’ mental health problems and perhaps even more keen in treating them. Conversely, providers in areas with sufficient mental health services may be more prone to expect their patients with mental health problems to seek treatment by specialists and thus do not prioritize mental health screening. While scarcity of mental health services is expanding in the U.S.42, further qualitative research can provide insights on how primary care providers in provider shortage areas and sufficiently covered areas perceive their professional roles when treating mental health problems.”

2. Adding to the same between-clinic heterogeneity issue, on p. 5 the authors write that “To control for the respective clinic in multivariable models, we also excluded patients who had encounters at various primary care clinics.” I am not sure I understand the logic for excluding these clients. In fact, an analysis that focuses on these patients in particular might be useful to uncover the puzzle of the between-clinic gaps.

Thank you for this suggestion. We have excluded those patients, as they likely switched providers during the study period, and thus data for a 14-month observation period at one provider is incomplete. A dataset with longer observational periods (e.g. 5 years) would be necessary to identify a sufficient number of patients switching clinics and to observe screening differences. However, such a dataset was not available (see also comment below). We are also unsure if a subsample of patients switching providers is representative of the entire patient population, as those patients may seek another primary care provider for distinct reasons (e.g. provision of specialized services at the new clinic, moving to another city, dissatisfaction with previous provider, etc.).

3. On p. 6, authors write that “The EMR system Epic prompts a PHQ-4 for every visit, however providers can skip entering results. Epic provides distinct fields to enter PHQ-4 responses and calculates sum scores automatically. Screened patients with a sum score ≥ 6, indicating clinically relevant depressive or anxious symptoms, were additionally asked to complete the more comprehensive PHQ-9 and GAD-7 as part of standard clinic procedure.”

Instead of the PHQ-4, do any patients receive these more comprehensive questionnaires or undergo some other mental health screening during their visits? If so, the outcome variable and analyses need to be amended.

Thank you for this comment. Indeed, there are a few patients with no documented PHQ-4 but a GAD-7 (n=62) or PHQ-9 (n=84). We have updated the outcome now defined as receiving either PHQ-4, PHQ-9, or GAD-7 in at least one encounter during the 14-month period. However, these changes are rather minor, e.g. 14-month prevalence for documented PHQ-4 is 79.712%, 14-month prevalence for any documented PHQ-4/PHQ-9/GAD-7 is 79.783%. We have updated these minor changes, where applicable throughout the manuscript, and also used this measure as an outcome for the multivariable analyses, and also amended the methods section accordingly.

“The primary outcome was receiving at least one PHQ-4 mental health screening (either PHQ-4/PHQ-9/GAD-7) during the study period.”

Additionally, of the patients who received the PHQ-4, examining how many scored 6 or over (or, if the scores are inaccessible, how many were given these more comprehensive questionnaires following the PHQ-4) could be an interesting outcome further elucidating the likelihood that different demographics’ mental illness goes undiagnosed.

While we have not recorded the score values of the respective instruments, 14.3% (n=14,444) of patients with a documented PHQ-4 screening during the study period were also screened with a GAD-7. Similarly, 10.6% (n=10,718) of PHQ-4 screened patients received an additional screening with the PHQ-9. We have added this information to the result section.

“On patient level, 79.8% (95% CI, 79.6 – 80.0) had a documentation of any PHQ-4/PHQ-9/GAD-7 screening during the 14-month period. 14.3% (n=14,444) of patients with a documented PHQ-4 screening during the study period were also screened with a GAD-7. Similarly, 10.6% (n=10,718) of PHQ-4 screened patients received an additional screening with the PHQ-9.”

Differences in comprehensive screening prevalence rates (PHQ-9/GAD-7) are indeed an important topic that needs further investigation. However, it is a prerequisite for those analyses that the sub-sample of PHQ-9/GAD-7 screened patients is representative of the non-PHQ-4 screened population. As we cannot ascertain this, the interpretation of results would be rather speculative.

4. Throughout the article, the use of the term “impact” (e.g. “The Impact of Sociodemographic Factors on Mental Health Screening”) should be changed to “association”. The evidence provided does not establish causality.

Thank you for this remark. We have changed the wording throughout the manuscript accordingly. 

5. Given the centrality of race to the framing of the article, I would include the Asian patient odds in the text, in addition to other racial groups (e.g., Abstract and p. 8). I believe the rates for women patients, too, are missing from p. 8.

Thank you for this remark. Due to word count limitations, we were only able to add aORs in the abstract for the factors with considerable effect sizes (aOR >1.5), but commented on this without reporting aORs. Given that we provide 2 models and all aORs are depicted in table 2, we decided not to repeat numbers but rather provide an overview of the important findings.

6. The discussion is currently too long. Precious word-count can be diverted to conducting and reporting further analyses, as suggested above. That being said, I am convinced by the point raised in the discussion that “limited mental health treatment capacities might reduce providers’ willingness to screen for mental health problems as they find themselves in a position where they cannot refer patients to specialists” (p. 11). I think this argument as well as the phenomenon of psychiatrist scarcity indicated by reference 42 could certainly be one of the mechanisms driving the between-clinic disparities this study observes. If indeed more rural clinics have lower rates of questionnaire administration, this would support this theory.

Thank you for this comment. We have restructured the entire discussion and made it more concise. Please see also responses above.

7. On p. 5, the authors mention that “Patients were excluded if they were not established patients at each respective clinic”. Why were these patients excluded? Was it necessary? I don’t believe Figure 1 properly reflects this exclusion.

Thank you for this comment. We have added a sentence defining “established patients”. 

“Established patients were all patients enrolled with a primary care provider of the respective clinic and having at least one encounter before 10/15/2021.”

The rationale for excluding non-established patients was that we did not want to mix in urgent-care patients or patients who receive regular primary care services with an established provider somewhere else. However, we agree that mental health screening prevalence among patients seeking urgent care is an important aspect that should be addressed in further investigations.

As an establishment with a provider was an á priori exclusion criteria, our honest broker excluded non-established patients before providing us the data. Thus, figure 1 cannot be amended.

More minor comments:

1. On p. 3, the sentence starting with “While there is a growing body of evidence” is missing some words.

Thank you for catching this error. We have rephrased the sentence.

2. On p. 8, authors report that “The 14-months screening prevalence rate was lowest among 34-year-olds (75.4%, n=1,940) and highest among 87-year-olds (88.4%, n=466).” It might be preferable to report (and statistically test) differences in questionnaire administration frequency on wider, 5- or 10-year age spans.

Thank you for this comment. The healthcare system underwent some restructuring and merging some years ago, thus unified EMR data covering a longer period was not available for this study. Despite these practical considerations, monitoring the impact of the release of screening recommendations or standard of care procedures in a longitudinal design was not objective in our study.

3. In the Discussion, on p. 10, the authors mention that “Our study showed that within a 14-month period, 79.7% of patients received a PHQ-4 screening with considerably lower rates for patients in their 20’s and 30’s.” This should be statistically tested.

Thank you for this remark. Figure 2 indicates 14-month PHQ-4 screening prevalence with corresponding 95% confidence intervals for respective ages. When error bars don’t overlap with the mean (dotted line), they are independent (p<0.05). When two error bars of two ages don’t overlap, the difference is also statistically significant on a p<0.05 level. It is arguable if further statistical tests would contribute to a better understanding, given the high number of included patients making rather small differences statistically significant.

I thank you for sharing this manuscript with me and wish you good luck in continuing your analysis of this interesting data.

---

## [Decision Letter · Decision Letter 1]

6 Feb 2024

PONE-D-23-16932R1Association between Sociodemographic Factors, Clinic Characteristics and Mental Health Screening Rates in Primary CarePLOS ONE

Dear Dr. Müller,

Thank you for submitting your manuscript to PLOS ONE. After careful consideration, we feel that it has merit but does not fully meet PLOS ONE’s publication criteria as it currently stands. Therefore, we invite you to submit a revised version of the manuscript that addresses the points raised during the review process.

Both initial reviewers have reviewed your revised manuscript again. Only reviewer 2 still has a few comments. I would like to ask you to revise your manuscript again with regard to these comments.

A minor comment from the second reviewer concerns the presentation of Table 2. In this context, I would also like to ask you to explain the difference between Model 1 and Model 2 in a footnote to Table 2. The additional adjustment for encounters is not immediately apparent due to the large number of factors examined. I would also like to remind you to check the references carefully. It does not comply with the PlosOne specifications (e.g. journal abbreviations are required, references with more than six authors should list the first six authors, references should be cited in parentheses, etc.).

We look forward to receiving your revised manuscript.

Kind regards,

Swaantje Wiarda Casjens

Academic Editor

PLOS ONE

Journal Requirements:

Reviewers' comments:

Reviewer's Responses to Questions

**Comments to the Author**

1. If the authors have adequately addressed your comments raised in a previous round of review and you feel that this manuscript is now acceptable for publication, you may indicate that here to bypass the “Comments to the Author” section, enter your conflict of interest statement in the “Confidential to Editor” section, and submit your "Accept" recommendation.

Reviewer #1: All comments have been addressed

Reviewer #2: (No Response)

2. Is the manuscript technically sound, and do the data support the conclusions?

Reviewer #1: Yes

Reviewer #2: Yes

3. Has the statistical analysis been performed appropriately and rigorously? 

Reviewer #1: Yes

Reviewer #2: Yes

4. Have the authors made all data underlying the findings in their manuscript fully available?

Reviewer #1: No

Reviewer #2: No

5. Is the manuscript presented in an intelligible fashion and written in standard English?

Reviewer #1: Yes

Reviewer #2: Yes

6. Review Comments to the Author

Reviewer #1: All comments addressed. Thank you in particular for addressing the concerns about age in the analysis which seem to have revealed interesting findings now reported and addressed in the discussion.

Reviewer #2: I am a returning Reviewer (no. 2). I commend the authors for responding to my prior comments and revising the article accordingly. I believe my prior concerns have been addressed. Reading the revised version, I do have one important comment remaining, as well a few more minor comments:

1. I thank the authors for now referring in the Discussion to the huge variability in screening rates they have observed between the clinics (ranging between 51.3% and 98.6%). To me, this still is one of the most important findings of the study, and I would encourage the authors to stress it even more. The existence of such heterogeneity in one, geographically bounded health system is surprising and alarming. Simultaneously, it may be encouraging, too. Namely, it can be seen an indication that beyond patient-level attributes that (as the authors show) contribute to under-screening and that may be more challenging for staff and practitioners to overcome (in terms of bias, cultural competence, etc.), the simple fact that some clinics screen all patients and others only one half indicates that perhaps even a more “blunt force” organizational, top-down, intervention ensuring screens are indeed routinely provided could have a hugely beneficial impact. The scale of this variance is quite shocking, raises interesting questions, and I would encourage the authors to dwell on it and its implications a bit more.

Minor comments:

2. The wording in the second paragraph of the Discussion, referring to reference no. 18 is slightly confusing. When the authors write “According to this study”, I suspect they may have meant “According to that study”. And so on. Further, I am not sure repeating the specific findings of that other study are of much importance. The key thing is that its findings on the demographic correlates of screening quite substantially differed from the current analysis, and that this may be attributable to its different methodology (pt. self-reports). This may be all the authors need to say about it.

3. I very much appreciated the analysis of the mental health professional shortage areas. I think it really adds depth to the discussion and I believe the article benefits from it quite considerably.

4. Lastly, I believe Table 2 is showing multivariate regression results, and so its columns should indicate aOR, and not OR.

Thank you for sharing with me this interesting research and for conducting what I believe to be a successful revision. Congratulations!

7. PLOS authors have the option to publish the peer review history of their article (what does this mean?). If published, this will include your full peer review and any attached files.

Reviewer #1: No

Reviewer #2: No

---

## [Author Response · Author response to Decision Letter 1]

26 Feb 2024

Reviewer #2:

I am a returning Reviewer (no. 2). I commend the authors for responding to my prior comments and revising the article accordingly. I believe my prior concerns have been addressed. Reading the revised version, I do have one important comment remaining, as well a few more minor comments:

Thank you again for your time and efforts in reviewing our manuscript.

1. I thank the authors for now referring in the Discussion to the huge variability in screening rates they have observed between the clinics (ranging between 51.3% and 98.6%). To me, this still is one of the most important findings of the study, and I would encourage the authors to stress it even more. The existence of such heterogeneity in one, geographically bounded health system is surprising and alarming. Simultaneously, it may be encouraging, too. Namely, it can be seen an indication that beyond patient-level attributes that (as the authors show) contribute to under-screening and that may be more challenging for staff and practitioners to overcome (in terms of bias, cultural competence, etc.), the simple fact that some clinics screen all patients and others only one half indicates that perhaps even a more “blunt force” organizational, top-down, intervention ensuring screens are indeed routinely provided could have a hugely beneficial impact. The scale of this variance is quite shocking, raises interesting questions, and I would encourage the authors to dwell on it and its implications a bit more.

Thank you for this important remark. The underutilization of screening instruments shown in our study stresses questions on how the implementation of this screening regime was carried out and how implementation strategies might be improved. We have now elaborated on this and added two paragraphs to each discussion and conclusion section: 

“Our study revealed significant screening rate variations among clinics, despite the health system having a longstanding standard procedure for mental health screening. This heterogeneity in one geographically bounded health system is surprising and concerning. Implementing mental health screening in the work routine was considerably more successful in some clinics than in others. This raises not only questions on structural and organizational barriers that may contribute to under-screening in those clinics but also indicates that a single top-down implementation effort may not be effective among some clinics which may require some more tailored and continuous approaches, e.g. constant monitoring and feedback, incentivizing of screening, highlighting the importance of screening in team meetings, education of office staff, reminder through EMR software, etc.”

“Furthermore, our study showed a considerable variety of mental health screening prevalence between clinics. This highlights the necessity for a continuous and tailored implementation effort to foster equitable mental health care.”

Minor comments:

2. The wording in the second paragraph of the Discussion, referring to reference no. 18 is slightly confusing. When the authors write “According to this study”, I suspect they may have meant “According to that study”. And so on. Further, I am not sure repeating the specific findings of that other study are of much importance. The key thing is that its findings on the demographic correlates of screening quite substantially differed from the current analysis, and that this may be attributable to its different methodology (pt. self-reports). This may be all the authors need to say about it.

Thank you for this comment. We have rephrased this section to make it more concise:

“While 14-month screening prevalence in our study was 79.8%, other researchers found considerably lower depression screening rates of 48.6% using a representative sample of US adults aged 35 or older.18 That study, however, used patient self-reported data and found other demographic factors correlated with mental health screening than in our study.”

3. I very much appreciated the analysis of the mental health professional shortage areas. I think it really adds depth to the discussion and I believe the article benefits from it quite considerably.

Thank you!

4. Lastly, I believe Table 2 is showing multivariate regression results, and so its columns should indicate aOR, and not OR.

Thank you for catching this. Indeed, it should read “aOR”. We have changed it.

Thank you for sharing with me this interesting research and for conducting what I believe to be a successful revision. Congratulations!

Thank you again for your helpful comments!

---

## [Editor Report · Decision Letter 2]

12 Mar 2024

Association between Sociodemographic Factors, Clinic Characteristics and Mental Health Screening Rates in Primary Care

PONE-D-23-16932R2

Dear Dr. Müller,

We’re pleased to inform you that your manuscript has been judged scientifically suitable for publication and will be formally accepted for publication once it meets all outstanding technical requirements.

Kind regards,

Swaantje Wiarda Casjens

Academic Editor

PLOS ONE
---

## [Editor Report · Acceptance letter]

19 Mar 2024

PONE-D-23-16932R2 

PLOS ONE

Dear Dr. Müller, 

I'm pleased to inform you that your manuscript has been deemed suitable for publication in PLOS ONE. Congratulations! Your manuscript is now being handed over to our production team.

Kind regards, 

on behalf of

Dr. Swaantje Wiarda Casjens 

Academic Editor

PLOS ONE